# Good Subjective Outcomes, Stable Knee and High Return to Sport after Tibial Eminence Avulsion Fracture in Children

**DOI:** 10.3390/children7100173

**Published:** 2020-10-09

**Authors:** Stefano Stallone, Filippo Selleri, Giovanni Trisolino, Alberto Grassi, Luca Macchiarola, Marina Magnani, Eleonora Olivotto, Stefano Zaffagnini, Stefano Stilli, Fabio Catani

**Affiliations:** 1Unit of Pediatrics Orthopedics and Traumatology, IRCCS Istituto Ortopedico Rizzoli, 40136 Bologna, Italy; giovanni.trisolino@ior.it (G.T.); marina.magnani@ior.it (M.M.); stefano.stilli@ior.it (S.S.); 2Struttura Complessa di Ortopedia e Traumatologia, University of Modena e Reggio Emilia, Policlinico di Modena, 41124 Modena, Italy; filipposelleri@gmail.com (F.S.); fabio.catani@unimore.it (F.C.); 3IIa Clinica Ortopedica e Traumatologica, IRCCS Istituto Ortopedico Rizzoli, 40136 Bologna, Italy; alberto.grassi@ior.it (A.G.); luca.macchiarola@ior.it (L.M.); stefano.zaffagnini@ior.it (S.Z.); 4RAMSES Laboratory, RIT Department, IRCCS Istituto Ortopedico Rizzoli, 40136 Bologna, Italy; eleonora.olivotto@ior.it

**Keywords:** tibial spine avulsion, tibial eminence fracture, pedi-IKDC, pedi-FABS, PAMI, return to sport

## Abstract

Avulsion fracture of the tibial spine (TSA) is uncommon in children, although its incidence is increasing with the earlier practice of competitive sport activities. This study aims to report mid to long term outcomes in children who sustained a TSA, with a special focus on a return to sport activities. Skeletally immature patients with a TSA, treated in two orthopedic hospitals, were evaluated for range of motion and knee laxity using KT1000, KiRA and Rolimeter. The pediatric International Knee Documentation Committee score (Pedi-IKDC) and the Hospital for Special Surgery pediatric Functional Activity Brief Scale (Pedi-FABS) questionnaires were recorded during the latest visit. Forty-two children were included. Twenty-six were treated nonoperatively and 16 underwent surgery. At a mean follow-up of 6.9 ± 3.6 years, 36 patients completed the questionnaires and 23 patients were tested with arthrometers. Among them, 96% had normal knee laxity. The Pedi-IKDC score averaged 96.4 ± 5.7 points, while the mean Pedi-FABS was 22.2 ± 5.9 points, without statistically significant differences between groups. Twenty-eight patients (78%) returned to their previous level of sport activity (eight amateur, 13 competitive, seven elite athletes). Eight patients (22%) quit sport, mostly because of re-injury fear. If properly treated, pediatric TSAs achieve a high rate of successful healing, with complete restoration of knee stability and an early return to sport activities.

## 1. Introduction

Paediatric avulsion fractures of the tibial spine (tibial spine avulsion: TSA) are uncommon fractures, with a quite low prevalence (three cases out of 100,000 children) [1], although the incidence is rising with the increasing and earlier practice of competitive sport activities [2].

TSAs are classified according to the Meyers and McKeever scoring system, as further modified by Zaricznyj, in four types [3,4]. So far, despite the fact that no complete evidence-based guidelines have been established yet, it is widely accepted that undisplaced fractures (type I) should be treated conservatively, while severely displaced fractures (types III and IV) require surgical reduction and fixation [5,6,7,8]. Borderline injuries (type II) show high rates of soft tissue entrapment, most commonly due to the inter-meniscal ligament or the meniscus itself. In these cases, the debate on non-operative versus operative management remains controversial. Both treatments show advantages and disadvantages. Among the latter, arthrofibrosis is the most frequently reported complication in surgical treatment, while higher rates of malunion or non-union have been described after conservative management. Residual knee laxity has been described after both surgical and nonsurgical management of TSA, but is not always symptomatic [5,6,8,9,10].

If properly treated, TSAs show favourably good long-term outcomes with a good functional recovery. However, there is insufficient evidence concerning factors affecting the return to sport activities after TSAs in children [11,12,13]. 

The aim of the present study was to retrospectively investigate the complications, subjective outcomes, joint laxity, and return to sport after either conservative or surgical management of the whole spectrum of TSA fractures in children. The hypothesis of the study was that similarly satisfactory results could be obtained with both surgical and conservative treatments in the case of proper indication based on TSA fracture patterns.

## 2. Materials and Methods

Ethics approval was sought and obtained from the local Ethical Committee (PG nr. 0002618 and nr. 0013211). The study was conducted in accordance with the Helsinki declaration and all patients gave informed consent in writing to participate. Parents provided written consent for the inclusion of the patients in this study, since all patients were minors (age less than 18) at the time of participation in the study.

A retrospective study was carried out on two tertiary referral centres for pediatric orthopedic, traumatology and sports medicine. A search of medical records was conducted to identify all patients treated for TSA in both hospitals between 2006 and 2017. Inclusion criteria were age <18 years, either conservative or surgical treatment and minimum follow-up of 2 years. Patients were excluded in the case of concomitant neuromuscular diseases, tibial plateau fractures, intrasubstance ACL tear, and multiple ligament injuries. TSA was diagnosed through anteroposterior and lateral plain radiographs. CT scans were obtained when radiographs were ambiguous either for TSA or tibial plateau fracture. MRIs were obtained, when possible, to better evaluate meniscal injury, osteochondral lesions, and ACL or collateral ligaments tearing. Considering the retrospective and multicentric nature of the study, which involved multiple clinicians and lasted more than 10 years, the treatment decision was not uniform. However, conservative or surgical treatment was proposed according to injury type and fragment dislocation, the presence of intact posterior hinge, and patients age. Moreover, the decision of the patient’s parents was taken into account.

### 2.1. Treatment

Conservative treatment consisted in an above-the-knee cast immobilization in full extension maintained for four to six weeks. Then, a hinged knee brace was used for 4 further weeks, allowing patients to start a rehabilitation program consisting in passive and active knee motion, muscle strengthening and gait recovery. 

Operative treatment consisted in either arthroscopic-assisted mini-open reduction and internal fixation (ORIF) or arthroscopic trans-osseous suture of the tibial spine. In both cases, standard arthroscopic portals (anterolateral and anteromedial) were performed and menisci, ligament structures and articular cartilage were assessed before reduction to avoid their interposition within the fracture site to minimize the risk of non-union. In the case of ORIF, a 2-cm medial para-patellar approach was performed, and the capsule was incised in line with the skin incision, exposing the fracture. A temporary fragment reduction was obtained with a K-wire, successively used as a guide to introduce a nonabsorbable, cannulated, partial threaded cancellous lag screw [Speyside Superscrew (Zimmer Biomet), Warsaw, IN, USA]. The screw placement was performed sparing the growth plate. In the case of arthroscopic transosseous suture, an Ethibond Excel [Ethicon (Johnson & Johnson), Miami, FL, USA] nonabsorbable suture was passed through the ACL substance, using a Caspari suture punch. Then, two tunnels were drilled within the proximal tibial epiphysis, directed toward the fracture bed. Sutures were retrieved through the tunnels and tied on the anterior tibial cortex using a pull-out technique. Finally, the range of movement, knee stability, and appropriate ACL tensioning were tested. Intraoperative fluoroscopy was performed to assure the correct reduction of the bony fragment and evaluate the correct positioning of the screw, when used.

Post-operatively, the knee was immobilized in a locked brace or a cast in 15° of flexion for 4–6 weeks. Full weight-bearing and progressive knee motion was allowed at brace removal. Full return to sport activities was allowed 4–6 months after injury according to clinical and radiological sign of bone healing, regardless surgical or conservative treatment.

### 2.2. Patients Evaluation

Patients were contacted and the Italian version [14] of the pediatric International Knee Documentation Committee score (Pedi-IKDC) (score of 0 to 100 with higher score equivalent to less pain and better function, and a minimally clinical important difference of 12 points) [15,16], and the Hospital for Special Surgery pediatric Functional Activity Brief Scale (Pedi-FABS) [17] (score 0 to 30 with higher score meaning better performance) were administered. Return to sport and further surgeries were inquired as well. Patients were also invited for a follow-up clinical visit to assess knee range of motion (ROM) and laxity. Anteroposterior laxity at 30° was evaluated measuring the side-to-side difference at manual maximum using the KT-1000 Arthrometer (MEDmetric Corp., San Diego, CA, USA) [18] while the anteroposterior laxity was measured using the Rolimeter [19]. Reliability and accuracy of these arthrometers was already assessed in previous studies [20,21,22] Knee laxity was considered normal with a side-to-side difference of <3 mm, nearly normal if 3–5 mm, and abnormal if >5 mm [20,23]. Rotatory laxity was measured with the Kira accelerometer device (I+, Italy) [24] while performing the Pivot-Shift manoeuvre; abnormal knee laxity was considered with values >0.8 mm/s^2^ [25]. All examinations were performed by a single experienced resident surgeon, after adequate training; the examinator was not blinded to the participants’ treatment. Medical records and charts were inquired to extract relevant information regarding the treatment course, fracture healing and complications. Arthrofibrosis was defined as 10-degree extension deficit or a 25-degree flexion loss [26].

### 2.3. Statistical Analysis

Continuous data were expressed as means, whereas categorical and ordinal data were expressed as absolute values and percentages. Normality was tested using the chi-square test for categorical variables and the Kolmogorov-Smirnov test for continuous variables. Differences in baseline and outcome characteristics between groups were tested using Fisher’s exact test for categorical variables and Student’s t-test for paired and unpaired data (normal distribution) or Mann–Whitney U-test and Wilcoxon signed-rank test (skewed distribution) for continuous variables. Exploratory univariable analyses with general linear models were performed to identify potential associations among baseline variables and outcomes. Statistical significance was set at *p* < 0.05 (2-tailed). All analyses were performed with SPSS v. 22.0 (SPSS, Chicago, IL, USA) and Microsoft Excel v. 16.30 (Microsoft Corporation, Redmond, WA, USA). Power analyses were performed using G*Power 3 (Heinrich-Heine University, Düsseldorf, Germany) [α error, 0.05; statistical power, 0.80; effect size, f2 = 0.6; number of predictors, 9]. The required sample size in this study was calculated as more than 36 participants.

## 3. Results

### 3.1. Patients Population

A total of 43 patients were treated for TSA fracture within the considered period. One patient was excluded due to a concomitant neuromuscular disorder. Finally, a total of 42 children (22 females, 20 males) with a mean age at trauma of 10.9 ± 3.2 years (range 5–16) were included in the study, with a mean follow-up of 6.9 ± 3.6 years (Figure 1). 

The TSA fractures, which occurred mainly during sport activities (81%) (Table 1), were distributed almost equally among the sexes (48% males, 52% females), with boys sustaining the injury at an older age when compared to girls (12.8 ± 2.6 vs. 9 ± 2.7; *p* < 0.0005).

According to the Meyers and McKeever system, 13 patients (31%) were classified as type I, 16 (37%) as type II, 12 (29%) as type III (eight IIIA and four IIIB), and four (3%) as type IV. Conservative treatment was used in 26 children (62%) while surgical treatment, either ORIF (n = 11) and suture (n = 5) in 16 children (38%) (Table 2). Most patients treated conservatively were of type I (62%). Type II (31%) and type III–IV (19%) fractures underwent non-operative treatment too, while only type II (50%) or type III (50%) fractures underwent surgical treatment. No demographic differences between patients treated either conservatively or surgically were detected (Table 2). However, when considering only patients with type II–IV fractures, patients treated conservatively were two years younger than those treated surgically.

### 3.2. Clinical Course, Fracture Healing and Complications

Among non-operatively treated patients, radiographic bone healing was achieved in 24/26 (92%) children at a mean of seven weeks (range 6–8) after injury. One patient with a type II fracture who was originally treated conservatively, presented further displacement at the 10-day follow-up visit, thus surgery was performed. For the purpose of the study, this case was considered a complication of the conservative treatment. He was however included in the operative cohort for the assessment of the final outcome. The second patient, a seven-year old girl with an acute type IIIA fracture, ended up in non-union since parents refused surgical treatment, despite recommendations. Delayed ACL reconstruction was proposed due to important knee laxity, but parents refused because the child only complained limited functional impairing and no mechanical symptoms up to four years of follow-up. None of the patients treated conservatively developed arthrofibrosis. 

Among patients treated surgically, with the numbers available, we did not find any significant difference between patients treated with ORIF and patients treated with suture. Radiographic bone healing was achieved at a mean follow-up of 7 weeks. Arthrofibrosis occurred in three out of 16 (19%) children, equally shared among ORIF (18%) and suture fixation (19%). These patients were managed with prolonged physical therapy for six to nine months from surgery and at the most recent follow-up no case of persistent knee stiffness was observed. No post-operative infection, progressive axial deformity, or leg length discrepancy was registered. Nine out of 11 patients (82%) treated with ORIF were re-operated for screw removal after an average of 13 months (range 5–21 months). 

### 3.3. Subjective Clinical Evaluation and Return to Sport

Thirty-six patients (84%) were available for subjective clinical evaluation at an average follow-up of 6.4 ± 3.3 years (range 2–12). Among the non-participants, five patients were definitely unreachable by phone, e-mail, or letter, while only one patient refused to be included in the study because he said he was unsatisfied with care. The average Pedi-IKDC was 96.4 ± 5.7, with no significant differences between surgically or conservatively treated patients (Table 3). Between 80% to 100% of patients reported the highest score in most of Pedi-IKDC “function” items, with the lowest performance reported for pain and instability during jumping or pivoting in patients who were managed nonoperatively (Figure 2). Only a minimal impairment was reported in the Pedi-IKDC “sport” items, mostly during jumping and landing (Figure 3). 

Twenty-eight patients (78%) returned to their previous level of sport activity (eight amateur, 13 competitive, seven elite level) (Table 4). Among the eight patients that quit sports, one did so because he started working and seven out of fear of a re-injury without complaining of any knee pain or instability. The average Pedi-FASB was 22.2 ± 5.9, with no significant differences among the two treatments (*p* = 0.32). Nearly 30–40% reported to be engaging in running, pivoting, and cutting activities more than four times per week (Figure 4). 

### 3.4. Objective Clinical Assessment

Among the assessed patients, 23 (64%) accepted to participate in objective clinical evaluation and laxity assessment (Table 3). The average side-to-side difference of manual maximum anterior displacement at 30° with KT-1000 was 0.4 ± 2.5 mm, with 20 patients (87%) having a value <3 mm and 2 patients (9%) between 3–5 mm. The only patient (4%) with a value >5 mm was the seven-year girl with type IIIA who had been treated conservatively following the parents’ will. The average side-to-side difference of anterior displacement at 90° with Rolimeter was 0.6 ± 1.3 mm. The average side-to-side difference of rotatory laxity during pivot-shift with Kira was 0.0 ± 0.8 mm/s^2^, with only two patients (9%) showing a value >0.8 mm/s^2^. A moderate correlation was found between Meyers and McKeever type and the side-to-side difference at KT1000 after adjustment for potential confounding factors (beta coefficient: 0.54, 95%CI: 0.16–2.67; *p* value = 0.03). 

Because of the limits imposed by the small patient cohort, no significant associations were found between clinical scores or laxity and patients characteristics.

## 4. Discussion

The most important finding of the present study is that successful outcomes, in terms of subjective scores, return to sport and objective knee laxity, can be achieved either with conservative or surgical treatment, based on TSA fracture type. Previous studies analyzed clinical and functional outcomes in children with TSA fractures. Janarv et al. [27] reported good results in 53 out of 61 patients (87%) assessed by Lysholm and Tegner scores. Perugia et al. [28] found a mean Tegner score of 6.6 (3–9) in 10 patients. Edmonds et al. [12] reported that 17 patients out of 18 were totally asymptomatic and able to return to sport at a preinjury level, but they did not use any specific tool for subjective or objective assessment of knee stability. Furthermore, all these studies used specific patients reported outcomes (PROs) measures that were originally designed for adult people, showing less validity, comprehensibility, and responsiveness in children [29,30]. Following the Paediatric ACL Monitoring Initiative (PAMI), we recently validated the Italian version of the pedi-IKDC and pedi-FABS scores. Both these scores were specifically designed for children and adolescents, having demonstrated good to excellent reliability and responsiveness [14,15,17].

The management of TSA still raises controversies. To date, it is widely accepted that undisplaced fractures (type I) are better managed conservatively, while major displacement (type III and IV) should require surgical repair [31]. The main concern remains the treatment of mild to moderate displacement in type II TSA fractures

A recent systematic review by Bugonovic reported several significant differences between non operatively treated and surgically treated TSA fractures, in terms of subjective feeling of instability, knee pain and limited ability to return to sport [9]. Furthermore, although several authors reported that conservative treatment had a high risk of developing knee laxity and instability [11,32], other authors reported no significant association between residual knee laxity and perceived instability or loss of self-confidence [25,33,34]. Overall, we observed satisfactory outcomes both in patients treated nonoperatively and in cases treated by surgical repair. Type II TSA fractures, treated either surgically or nonoperatively, achieved excellent results, in terms of knee symptoms, function, sport activity and objective knee stability. Also type III and IV TSA fractures exhibited satisfactory results in terms of knee function and stability, although most of the patients who sustained a type III or IV TSA fractures and were treated conservatively reported some residual knee pain during daily activities. While these findings were not statistically significant with the numbers available, we suggested surgical repair in case of major displacements (type III and IV), especially in children who want to return to sport activity, while minor displacements (type II) should be evaluated case by case considering the age of patients, since younger children have higher healing and bone remodeling potential, and the psychological motivation and the pre-injury level of sport activity.

Concerning the type of fixation, we did not find any significant difference between bone suture and screw fixation, both in terms of clinical and radiographic outcomes. This finding could be affected by the small cohort. Nonetheless other studies with large cohorts showed no differences regarding postsurgical arthrofibrosis, instability, range of motion and return to sport [10]. Furthermore, in our practice, we usually plan prophylactical hardware removal in children, which is not needed in the case of bone suture. This could be a possible reason to prefer the latter.

Despite objective knee stability was restored in more than 90% of cases and 77% children resumed sport at a preinjury level of performance (including 20% elite athletes), almost 20% of children quit sport activity because of referred pain and fear of reinjury. We cannot distinguish if pain was determined by mechanical problems or psychological issues, since there were no differences, both in terms of knee laxity and knee motion between children who continued sport activity and children that abandoned sport. Psychological aspects should be better investigated in very young athletes, since anxiety, kinesiophobia, and loss of self-confidence are well-known factors that may lead to sport retirement after ACL reconstruction [35,36]. Moreover, considering the controversial literature and the limited case series, multicentric initiatives -eventually under the umbrella of Scientific Societies- should be encouraged to monitor TSA outcomes and identify the optimal management.

The present study has limitations. The cohort is limited in numbers and heterogeneous in treatments, thus difficult to make comparisons and draw any conclusions. The retrospective design and lack of randomization prevent any speculation about cause and effect relationships among factors potentially affecting the clinical outcomes. PROMs and instrumental knee laxity were assessed only at the latest follow-up, thus not allowing subject assessment of knee symptoms and stability. Furthermore, the investigators were not blinded to the participants’ treatment. One patient in the conservative group changed treatment method from conservation to operation, which may produce bias and influence final results. However, effect of this bias would not be substantial. Despite our collaborative initiative involving two high-volume orthopaedic hospitals, our study reached 80% power to detect large effect sizes (f2 = 0.6) but just 45% power to detect medium effects sizes (f2 = 0.3), including nine potential predictors of the final outcome. Consequently, our study is underpowered to detect medium or small effect sizes. A multicentre prospective data collection could overcome this issue, possibly extending the activity of the PAMI registry also to TSA fractures. Furthermore, most patients in our cohort showed complete stability and excellent long-term outcomes. This could affect the responsiveness of our tests, because of an important ceiling effect, limiting the differences between patients.

Finally, only 55% patients accepted to be tested with arthrometers. This is an important drawback in our analysis but is a common issue even in other studies investigating this uncommon injury, where loss to follow-up ranged from 17% to 76% [10,11,12,37]. However, we believe that no significantly different knee laxity is to be detected between children who were tested and children who were not tested, since we did not find any difference in terms of clinical outcomes.

## 5. Conclusions

If properly treated, pediatric avulsion fractures of the tibial eminence achieve a high rate of successful healing, with complete restoration of knee stability and in most cases of function, and early return to sport activities at a pre-injury level. Despite this, many children stopped playing sports, mainly because of fear of re-injury. 

## Figures and Tables

**Figure 1 children-07-00173-f001:**
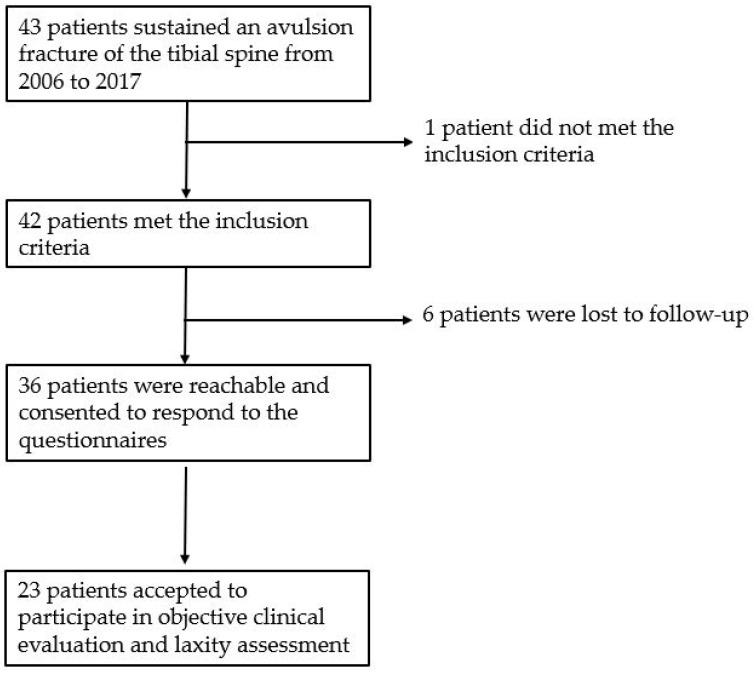
Flow-chart of patients who were lost to follow-up.

**Figure 2 children-07-00173-f002:**
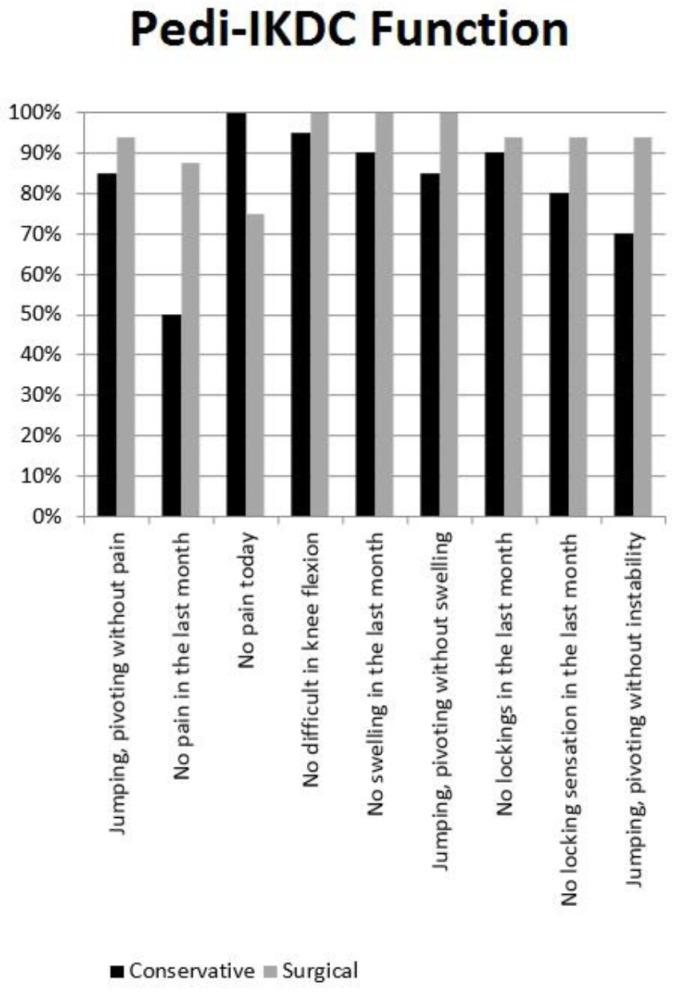
Pedi-IKDC Function.

**Figure 3 children-07-00173-f003:**
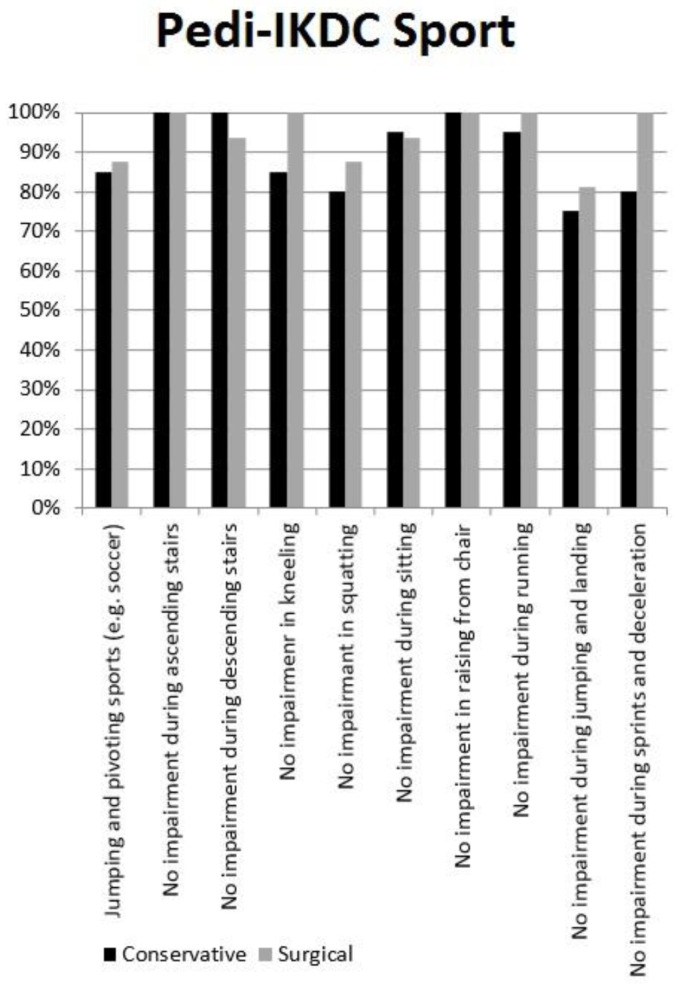
Pedi-IKDC Sport.

**Figure 4 children-07-00173-f004:**
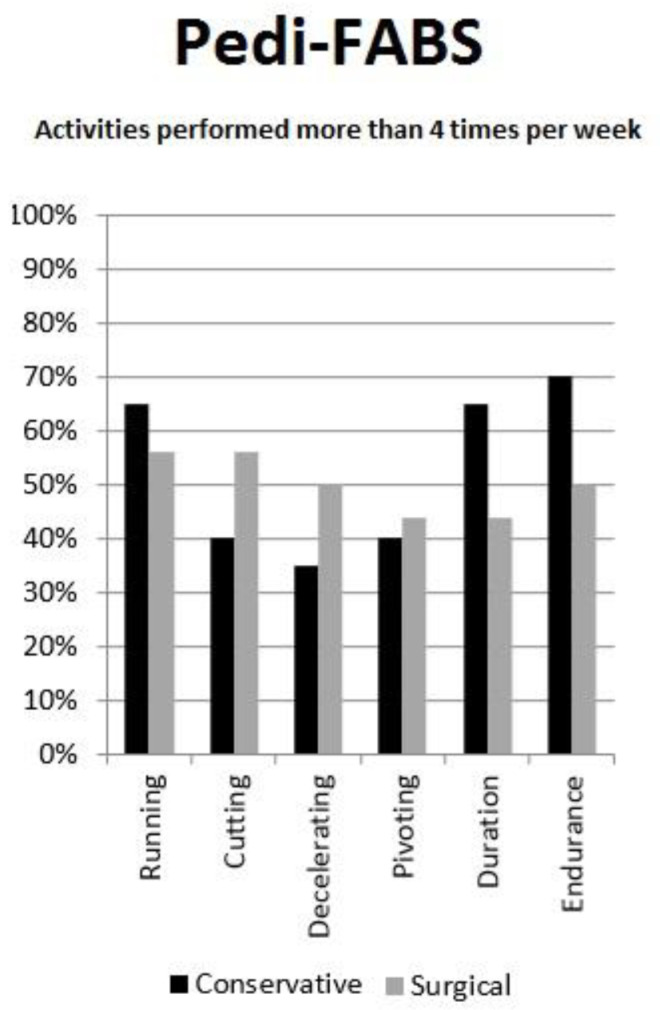
Pedi-FABS.

**Table 1 children-07-00173-t001:** Causes of fracture.

Number of Patients	Type of Trauma		
**34**		15	Ski/Snowboarding
		8	Motocross
	**Sport**	4	Soccer
		3	Basket
		3	Biking
		1	Volleyball
**3**	**Car accident**		
**5**	**Fall**		

**Table 2 children-07-00173-t002:** Patients characteristics and baseline variables.

	Non-Operative	Operative	Total	*p*
**N° of patients**	26	16	42	
**Hospital A/Hospital B Ratio**	17/9	5/11	22/20	0.055
**Male/Female**	10/16	10/6	20/22	0.17
**Left/Right**	7/19	11/5	18/24	0.01
**Meyers and McKeever class. (I/II/IIIA/IIIB/IV)**	13/8/3/1/1	0/8/5/3/0	13/16/8/4/1	0.005
**Age at trauma**	10.6 ± 3.6	11.3 ± 2.5	10.8 ± 3.2	0.46
**Follow-up (years)**	8.0 ± 3.7	5.9 ± 2.6	6.9 ± 3.6	0.02
**Age at last follow-up**	18.4 ± 5.0	16.6 ± 4.1	17.7 ± 4.8	0.24

All results are expressed as variables and as mean ± Standard Deviation for continue variables. Statistical significance was set at *p* < 0.05.

**Table 3 children-07-00173-t003:** Patient-reported questionnaires outcomes and instrumental knee laxity measurements.

OUTCOME VARIABLE	Non-Operative	Operative	Total	*p* Value
**PROs Questionnaires**				
**Number of patients**	20	16	36	
**Pedi-IKDC**	95.4 ± 6.6	97.8 ± 4.0	96.4 ± 5.7	0.58
**Pedi FABS**	22.1 ± 5.6	22.5 ± 6.8	22.2 ± 5.9	0.32
**Instrumental Knee laxity**				
**Number of patients**	13	10	23	
**KT 1000**	0.15 ± 3.05	0.8 ± 1.68	0.4 ± 2.46	0.16
**Rolimeter**	0.46 ± 1.12	0.7 ± 1.63	0.6 ± 1.32	0.45
**KiRA**	0.14 ± 0.66	−0.13 ± 0.89	0.05 ± 0.74	0.37

All results are expressed as variables and as mean ± Standard Deviation for continue variables. Statistical significance was set at *p* < 0.05. PROs = Patients Related Outcomes; Pedi-IKDC: pediatric International Knee Documentation Committee score; Pedi-FABS: Hospital for Special Surgery pediatric Functional Activity Brief Scale.

**Table 4 children-07-00173-t004:** Type and level of sport of patients after TSA fracture.

Level of Sport	Male	Female	TOT
**Elite**	Soccer × 2 Motocross × 2 Ski	Athletics Baseball Dance/Ballet	8
**Competitive**	Soccer × 2 Basket × 2 Motocross Snowboard Swim	Ski/Snowboard × 3 Dance/Ballet × 2 Horse Riding	13
**Amateur**	Soccer Basket	Athletics × 2 Swim × 2 Dance/Ballet	7
**TOT**	14	14	28

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
