# Peer review of "Good Subjective Outcomes, Stable Knee and High Return to Sport after Tibial Eminence Avulsion Fracture in Children"

_children, 2020, doi:10.3390/children7100173_

Round 1
Reviewer 1 Report
An interesting, well written retrospective study with a focus on a novel outcome (return to sport) for an uncommon injury. The use of paediatric specific knee scores is also interesting.
A few minor points:
1) You state on lines 67-69 that the treatment algorithm was not standardised, which is understandable given the nature of the study. However, you go on to describe in detail the conservative and surgical treatment approaches. So, do you mean the decision to operate or not was not standardised? Or that the treatments themselves were not standardised? This could be clearer.
2) You describe two operative approaches ORIF v arthroscopic. Have you explored if there any differences in outcomes or complications between these different operative approaches?
3) I am unclear what an "intermittent brace" is (Line 74)? Is this like a hinged knee brace which restricts flexion/exension? This could be clearer.
4) In section 2.2 you describe how patient evaluation was undertaken. What you do not mention is where or who this was performed by. Perhaps an idea of their role/qualification to undertake this evaluation would be useful. Also, was it the same person assessing all patients and at a single centre with the same equipment? Was the assessor blinded to treatment group or not? I suspect sliding would not be possible but this should be stated.
5) Were there any differences in patients that consented to evaluation or not? For example were all patients who presented to follow up from a single centre? Or with a particular injury grade? Possibly only those who are happy with their outcome attended for evaluation?
6) You use the word 'quitted' several times. I think the better English would be to use the word 'quit'
7) Line 130 'mean age at trauma' I suspect you meant to say 10.9 +/- 3.2 years? The word years is missing
8) Line 141 you quote a p-value of 0.005, what is this for? It seems an unnecessary place to perform a statistical test
9) Figure 1 - The X-axis could be clearer. If this is showing injury grade by year then simply put the year on the x-axis. I'm also unclear what you are trying to demonstrate with this figure? That there is no correlation between treatment choices over time? It doesn't seem to add much to the article.
10) Line 167 you mention screw removal was performed in 9 of 11 patients. Why was this undertaken? Due to stiffness or simply planned at time of the index procedure? Linking to my earlier comment about are there any differences between surgical treatment groups, the need for re-operation for screw removal could be a benefit of suture techniques, if the outcomes are no different.
11) I like the use of paediatric specific knee scores. You demonstrate no statistically significant difference in scores but what is the minimal clinically important difference for these scores? Similarly for your measures of laxity you give very small average measurements i.e. 0.4mm, 0.6mm. How accurate are the devices being used? What amount of displacement or laxity is considered clinically important or likely to cause symptoms?
Overall I think this is a good paper but could be improved prior to publication.
Author Response
We thank the reviewer for his/her valuable comments and suggestions.
In the attachment you can find a point-by-point answer to your comments.

Reviewer 2 Report
We read with great interest this manuscript and have a few remarks:- Could you add a flowchart explaining which patients in the initial sample were lost to follow-up?
- For patients who underwent preoperative or initial MRI, have you found any entrapment of soft tissue such as the anterior horn of the meniscus or the inter-meniscal ligament?- Is it possible to do the analysis excluding type I fractures. Would it be relevant to compare only the patients of stages II / IIIA / IIIB / IV in the operated / non-operated groups. We would be free from the "good" results associated with a less severe fracture pattern. Their presence leads to a lack of homogeneity of the groups.
- How do you explain that 5 fractures classified as IIIa, IIIb and IV were treated conservatively when these types of fractures are the fractures to be reduced and fixed.
- Why did patients who responded to the Pedi-IKDC test not take the laximetry tests?
- In the discussion, line 235 “Also type III and IV TSA fractures exhibited satisfactory results in terms of knee function and stability, although most of the patients who sustained a type III or IV TSA fractures and were treated conservatively reported some residual knee pain during daily activities. " Does this mean that patients with daily pain are satisfied?
- Did some patients in the conservative group need subsequent surgery? intra-articular callus resection or ligamentoplasty?
Author Response

(The authors gave the same response as above.)
